# Learnability Lock: Authorized Learnability Control Through Adversarial Invertible Transformations

**Weiqi Peng**
Yale University
`weiqi.peng@yale.edu`

**Jinghui Chen**
Pennsylvania State University
`jzc5917@psu.edu`

## Abstract

Owing much to the revolution of information technology, the recent progress of deep learning benefits incredibly from the vastly enhanced access to data available in various digital formats. However, in certain scenarios, people may not want their data being used for training commercial models and thus studied how to attack the learnability of deep learning models. Previous works on learnability attack only consider the goal of preventing unauthorized exploitation on the specific dataset but not the process of restoring the learnability for authorized cases. To tackle this issue, this paper introduces and investigates a new concept called "learnability lock" for controlling the model's learnability on a specific dataset with a special key. In particular, we propose adversarial invertible transformation, that can be viewed as a mapping from image to image, to slightly modify data samples so that they become "unlearnable" by machine learning models with negligible loss of visual features. Meanwhile, one can unlock the learnability of the dataset and train models normally using the corresponding key. The proposed learnability lock leverages class-wise perturbation that applies a universal transformation function on data samples of the same label. This ensures that the learnability can be easily restored with a simple inverse transformation while remaining difficult to be detected or reverse-engineered. We empirically demonstrate the success and practicability of our method on visual classification tasks.

## 1 Introduction

Recent progress in deep learning empowers the application of artificial intelligence in various domains such as computer vision (He et al., 2016), speech recognition (Hinton et al., 2012), natural language processing (Devlin et al., 2018) and etc. While the massive amounts of publicly available data lead to the successful AI systems in practice, one major concern is that some of them may be illegally exploited without authorization. In fact, many commercial models nowadays are trained with unconsciously collected personal data from the Internet, which raises people's awareness on the unauthorized data exploitation for commercial or even illegal purposes.

For people who share their data online (e.g., upload selfie photos to social networks), there is not much they can do to prevent their data from being illegally collected for commercial use (e.g. training commercial machine learning models). To address such concern, recent works propose *Learnability Attack* (Fowl et al., 2021a; Huang et al., 2021; Fowl et al., 2021b), which adds invisible perturbations to the original data to make it "unlearnable" such that the model trained on the perturbed dataset has awfully low prediction accuracy. Specifically, these methods generate either sample-wise or class-wise adversarial perturbations to prevent machine learning models from learning meaningful information from the data. This task is similar to traditional data poisoning attack, except that there is an additional requirement that the perturbation noise should be invisible and hard-to-notice by human eyes. And when the model trained on the perturbed dataset is bad enough, it won't even be considered as a realistic inference model by the unauthorized user for any purposes.

However, the previous works on learnability attack only consider the goal of making the perturbed dataset "unlearnable" to unauthorized users, while in certain scenarios, we also want the authorized

users to access and retrieve the clean data from the unlearnable counterparts. For example, photographers or artists may want to add unlearnable noise to their work before sharing it online to protect it from being illegally collected to train commercial models. Yet they also want authorized users (e.g., people who paid for the copyrights) to obtain access to their original work. In this regard, the existing methods (Fowl et al., 2021a; Huang et al., 2021; Fowl et al., 2021b) have the following major drawbacks: (1) sample-wise perturbation does not allow authorized clients to recover to original data for regular use, i.e., restore the learnability; (2) class-wise perturbation can be easily detected or reverse engineered since it applies the same perturbation for all data in one class; (3) the unlearnable performances are significantly worse for adversarial training based learning algorithms, making the perturbed data, to a certain extent, "learnable" again.

To tackle these issues, we introduce the novel concept of *Learnability Lock*, a learnability control technique to perturb the original data with adversarial invertible transformations. To meet our goal, a DNN model trained on the perturbed dataset should have a reasonably bad performance for various training schedules such that none of the trained models can be effectively deployed for an AI system. The perturbation noises should also be both invisible and diverse against visual inspections. In addition, the authorized users should be able to use the correct key to retrieve the original data and restore their learnability.

In particular, we creatively adopt a class-wise invertible functional perturbation where one transformation is applied to each class as a mapping from images to images. In this sense, the key for the learnability control process is defined as the parameters of the transformation function. For an authorized user, the inverse transformation function can be easily computed with the given key, and thus retrieving the original clean dataset. We focus our method on class-wise noise rather than sample-wise noise because we want the key, i.e., parameters of the transformation function, to be lightweight and easily transferable. Note that different from traditional additive noises (Fowl et al., 2021a; Huang et al., 2021; Fowl et al., 2021b), even the same transformation function would produce different perturbations on varying samples. Therefore, it is still difficult for an attacker to reverse-engineer the perturbation patterns as a counter-measure.

We summarize our contributions as follows:

- We introduce a novel concept *Learnability Lock*, that can be applied to the scenario of learnability control, where only the authorized clients can access the key in order to train their models on the retrieved clean dataset while unauthorized clients cannot learn meaningful information from the released data that appears normal visually.
- We creatively apply invertible functional perturbation for crafting unlearnable examples, and propose two efficient functional perturbation algorithms. The functional perturbation is more suited to crafting class-wise noises while allowing different noise patterns across samples to avoid being detected or reverse engineered.
- We empirically demonstrate that our pipeline works on common image datasets and is more robust to defensive techniques such as adversarial training than prior additive perturbation methods (Fowl et al., 2021a; Huang et al., 2021; Fowl et al., 2021b).

## 2 RELATED WORK

**Adversarial Attack** One closely related topic is adversarial attack (Goodfellow et al., 2015; Papernot et al., 2017). Earlier studies have shown that deep neural networks can be deceived by small adversarially designed perturbations (Szegedy et al., 2013; Madry et al., 2017). Later on, various attacks (Carlini & Wagner, 2017; Chen et al., 2017; Ilyas et al., 2018; Athalye et al., 2018; Chen et al., 2020; Moon et al., 2019; Croce & Hein, 2020; Tashiro et al., 2020; Andriushchenko et al., 2020; Chen & Gu, 2020) were also proposed for different settings or stronger performances. Laidlaw & Feizi (2019) proposes a functional adversarial attack that applies a unique color mapping function for crafting functional perturbation patterns. It is proved in Laidlaw & Feizi (2019) that even a simple linear transformation can introduce more complex features than additive noises. On the defensive side, adversarial training has been shown as the most efficient technique for learning robust features minimally affected by the adversarial noises (Madry et al., 2018; Zhang et al., 2019a).

**Data Poisoning** Different from adversarial evasion attacks which directly evade a model's predictions, data poisoning focuses on manipulating samples at training time. Under this setting, the

attacker injects bad data samples into the training pipeline so that whatever pattern the model learns becomes useless. Previous works have shown that DNNs (Muñoz-González et al., 2017) as well as traditional machine learning algorithms, such as SVM (Biggio et al., 2012), are vulnerable to poisoning attacks (Shafahi et al., 2018; Koh et al., 2018). Recent progress involves using gradient matching and meta-learning inspired methods to solve the noise crafting problem (Geiping et al., 2020; Huang et al., 2020). One special type of data poisoning is backdoor attacks, which aim to inject falsely labeled training samples with a stealthy trigger pattern (Gu et al., 2017; Qiao et al., 2019). During the inference process, inputs with the same trigger pattern shall cause the model to predict incorrect labels. While poisoning methods can potentially be used to prevent malicious data exploitation, the practicability is limited as the poisoned samples appear distinguishable to clean samples (Yang et al., 2017).

**Learnability Attack** Learnability attack methods emerge recently in the ML community with a similar purpose as data poisoning attacks (to reduce model's prediction accuracy) but with an extra requirement to preserve the visual features of the original data. Huang et al. (2021) firstly present a form of error-minimizing noise, effective both sample-wise and class-wise, that forces a deep learning model to learn useless features and thus behave badly on clean samples. Specifically, the noise crafting process involves solving a bi-level objective through projected gradient descent (Madry et al., 2018). Fowl et al. (2021b) resolves a similar problem using gradient matching technique. In another work, they note that adversarial attacks can also be used for data poisoning (Fowl et al., 2021a). In comparison, our method concerns a novel target as "learnability control", where authorized user is granted access to retrieve the clean data and train their model on it.

## 3 METHOD

### 3.1 PROBLEM DEFINITION

We focus on the scenario where the data for releasing is intentionally poisoned by the data holder to lock its learnability. In particular, a model trained on the learnability locked dataset by unauthorized users should have awfully low performances on the standard test cases. On the other hand, authorized clients with the correct keys can unlock the learnability of the dataset and train useful models. In designing a secure learnability control approach for our needs, we assume the data holder has full access to the clean data, but cannot interfere with the training procedure after the dataset is released. We also assume that the dataset to be released is fully labeled. Let's denote the original clean dataset with $n$ samples as $D_c = \{(\mathbf{x}_i, y_i)\}_{i=1}^n$. Our goal is to develop a method that succeeds in the following tasks:

**A.** Given $D_c$, craft the learnability-locked dataset $D_p = \{(\mathbf{x}_i', y_i)\}_{i=1}^n$ with a special key $\psi$.

**B.** Given $D_p$ and the corresponding key $\psi$, recover the clean dataset as $D_{\widetilde{c}}$, which restores the learnability of $D_c$.

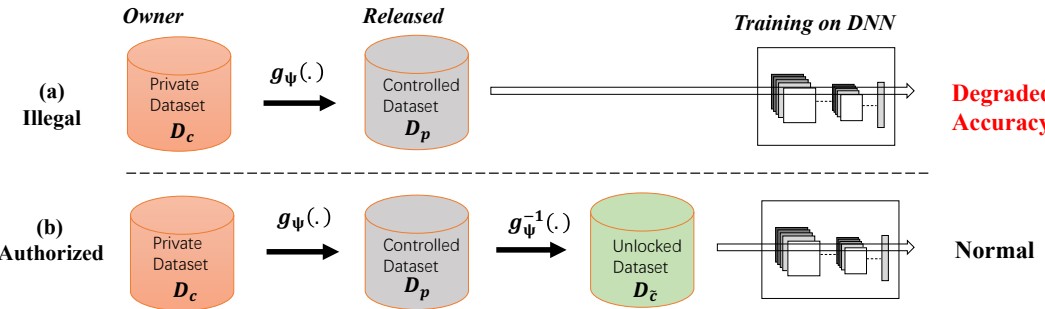

Figure 1: Authorized learnability control pipeline, with $g_\psi$ being a perturbation function where $\psi$ serves as the key for the learnability lock. (a) demonstrates the case of an illegal exploitation attempt where the unauthorized client trains a model on the learnability-locked dataset; In (b) an authorized user can unlock the protected dataset with the inverse transformations and train a normal model.

Now we design a novel method that introduces functional perturbations to generate learnability locked data samples. For simplicity, let's consider the common supervised image classification

task with the goal of training a K-class classifier $f_{\boldsymbol{\theta}} : \mathcal{X} \to \mathcal{Y}$ with a labeled dataset. With a little abuse of notations, we define a set of $K$ perturbation functions $\{g^{(1)}, ..., g^{(K)}\}$ for each class $y \in \{1, ..., K\}$. Each perturbation function $g^{(y)} : \mathcal{X} \to \mathcal{X}$ maps the original data $\mathbf{x}$ to a new one $\mathbf{x}'$, through an invertible transformation while the changes remain unnoticeable, i.e., $\mathbf{x}'^{(y)} = g^{(y)}(\mathbf{x})$ and $\|\mathbf{x}' - \mathbf{x}\|_{\infty} \le \epsilon$. Let's use $\boldsymbol{\psi}$ to denote the parameters of the transformation function, which also served as the key for unlocking the learnability of the dataset. To solve for the most effective transformation function for locking the learnability, we formulate the problem as a min-min bi-level optimization problem for each class $y$:

$$\underset{\boldsymbol{\theta}}{\operatorname{argmin}} \, \mathbb{E}_{(\mathbf{x},y) \sim \mathcal{D}_c} \left[ \min_{\boldsymbol{\psi}} \mathcal{L} \left( f_{\boldsymbol{\theta}}(g_{\boldsymbol{\psi}}^{(y)}(\mathbf{x})), y \right) \right] \quad \text{s.t. } \|g_{\boldsymbol{\psi}}^{(y)}(\mathbf{x}) - \mathbf{x}\|_{\infty} \le \epsilon, \tag{3.1}$$

where $\boldsymbol{\theta}$, the model for noise generation, is trained from scratch and $\boldsymbol{\psi}$, the parameters of the perturbation function, is also initialized as identical mapping.

Intuitively, the inner minimization will generate a perturbation such that the perturbed data is well fit the current model. Thus in the outer minimization task, the model is tricked into believing that the current parameters are perfect and there is little left to be learned. Therefore, such perturbations can make the model fail in learning useful features from data. Finally, training on such perturbed dataset will lead to a model that heavily relies on the perturbation noise. And when testing the model on clean samples, it cannot judge based on the noise pattern and thus makes wrong predictions. (3.1) can be solved via an alternating strategy: first solving the inner minimization problem to obtain the best $\boldsymbol{\psi}$ that minimizes the training loss, then solving the outer minimization for better $\boldsymbol{\theta}$. Ideally, if (3.1) can be well optimized, the perturbations generated by $g_{\boldsymbol{\psi}}^{(y)}(\cdot)$ should fulfill task A of crafting unlearnable samples.

Compared with Huang et al. (2021), the main difference is that we define the inner optimization as solving the transformation function parameters $\boldsymbol{\psi}$ instead of a set of fixed perturbation vectors. This brings us unique challenges in designing the form of the perturbation functions $g_{\boldsymbol{\psi}}^{(y)}(\cdot)$. Specifically, to fulfill task B, the perturbation function $g$ is required to be invertible (i.e. $g^{-1}$ exists), which means it maps every input to a unique output and vice versa. Moreover, the magnitude of perturbation should be bounded so that it will not affect normal use, i.e., $\|g_{\boldsymbol{\psi}}^{(y)}(\mathbf{x}) - \mathbf{x}\|_{\infty} \le \epsilon$. This constraint may seem easy for generating adversarial examples but not quite obvious for building invertible transformations. Finally, it requires a careful balance between the transformation's parameter size and its expressive power: it should be complicated enough to provide various perturbations for solving (3.1) and also be relatively lightweight for efficiently transferring to the authorized clients.

In the following section, we present two plausible perturbation functions that could satisfy the aforementioned properties. One is based on linear transformation and the other one is convolution-based nonlinear transformation.

### 3.1.1 LINEAR FUNCTIONAL PERTURBATION

We start by considering a linear transformation function where $g_{\boldsymbol{\psi}}^{(y)}(\cdot)$ consists of simple element-wise linear mappings over input data $\mathbf{x}_i \in \mathbb{R}^d$ drawn from $D_c$. We define the linear weights $\mathbf{W} = \{\mathbf{w}^{(1)}, ..., \mathbf{w}^{(K)}\}$, with each $\mathbf{w}^{(y)} \in \mathbb{R}^d$, and biases $\mathbf{B} = \{\mathbf{b}^{(1)}, ..., \mathbf{b}^{(K)}\}$, with each $\mathbf{b}^{(y)} \in \mathbb{R}^d$. We denote the overall parameters $\boldsymbol{\psi} = \{\mathbf{W}, \mathbf{B}\}$. For each $(\mathbf{x}_i, y_i) \in D_c$, we leverage an invertible transformation as:

$$\mathbf{x}_i' = g_{\boldsymbol{\psi}}^{(y_i)}(\mathbf{x}_i) = \mathbf{w}^{(y_i)} \odot \mathbf{x}_i + \mathbf{b}^{(y_i)}, \tag{3.2}$$

where $\odot$ denotes element-wise multiplication. Note that the above transformation is class-wise, where weights $\mathbf{w}$ and $\mathbf{b}$ are universal for each class label. Furthermore, for each index $j \in [d]$ we restrict $[\mathbf{w}^{(y)}]_j \in [1 - \epsilon/2, 1 + \epsilon/2]$, and $[\mathbf{b}^{(y)}]_j \in [-\epsilon/2, \epsilon/2]$. This ensures that the perturbation on each sample shall be bounded by:

$$\|\mathbf{x}_i' - \mathbf{x}_i\|_{\infty} = \max_{j \in [d]} \left| [\mathbf{w}^{(y)}]_j [\mathbf{x}_i]_j + [\mathbf{b}^{(y)}]_j - [\mathbf{x}_i]_j \right| \le \frac{\epsilon}{2} \max_{j \in [d]} |[\mathbf{x}_i]_j| + \frac{\epsilon}{2} \le \epsilon,$$

where the last inequality is due to $0 \le [\mathbf{x}_i]_j \le 1$ in the image classification task. Therefore, the linear function implicitly enforces the perturbation to satisfy the $\epsilon$ limit in $L_{\infty}$ distance. The whole crafting process is summarized in Algorithm 1.

---

**Algorithm 1** Learnability Locking (Linear)

---

1: **Inputs:** Clean dataset $D_c = \{(\mathbf{x}_i, y_i)\}_{i=1}^n$, crafting model $f$ with parameters $\boldsymbol{\theta}$, all $\mathbf{w}_0 = \mathbf{1}$ and $\mathbf{b}_0 = \mathbf{0}$, perturbation bound $\epsilon$, outer optimization step $I$, inner optimization step $J$, error rate $\lambda$, loss function $L$
2: **while** error rate $< \lambda$ **do**
3:     $D_p \leftarrow$ empty dataset
4:     **for** $(\mathbf{x}_i, y_i)$ **in** $D_c$ **do**
5:         $\mathbf{x}'_i = \mathbf{w}^{(y_i)} \odot \mathbf{x}_i + \mathbf{b}^{(y_i)}$
6:         Add $(\mathbf{x}'_i, y_i)$ to $D_p$
7:     **end for**
8:     **repeat** $I$ steps:                                   ▷ outer optimization over $\boldsymbol{\theta}$
9:         $(\mathbf{x}'_k, y_k) \leftarrow$ Sample a new batch from $D_p$
10:        Optimize $\boldsymbol{\theta}$ over $\mathcal{L}(f_\theta(\mathbf{w}_t^{(y_k)} \odot \mathbf{x}'_k + \mathbf{b}_t^{(y_k)}), y_k)$ by SGD
11:     **repeat** $J$ steps:                                ▷ inner optimization over $\boldsymbol{\psi}$
12:         **for** $(\mathbf{x}'_i, y_i)$ **in** $D_p$ **do**
13:            Optimize $\mathbf{W}$ and $\mathbf{B}$ by (3.3) and (3.4)
14:            $\mathbf{W} \leftarrow$ Clip$(\mathbf{W}, 1 - \epsilon/2, 1 + \epsilon/2)$, and $\mathbf{B} \leftarrow$ Clip$(\mathbf{B}, -\epsilon/2, +\epsilon/2)$
15:         **end for**
16:     error rate $\leftarrow$ Evaluate $f_{\boldsymbol{\theta}}$ over $D_p$
17: **end while**
18: **Output** Poisoned dataset $D_p$, key $\boldsymbol{\psi} = (\mathbf{W}, \mathbf{B})$

---

**Algorithm 2** Learnability Unlocking (Linear)

---

1: **Inputs:** Poisoned dataset $D_p = \{(\mathbf{x}'_i, y_i)\}_{i=1}^n$, CNN parameters $\boldsymbol{\theta}$, $\mathbf{W}$ and $\mathbf{B}$, perturbation bound $\epsilon$
2:  $D_{\widetilde{c}} \leftarrow$ empty list
3: **for** each $\mathbf{x}'_i, y_i$ in $D_p$ **do**
4:     $\mathbf{x}_i = (\mathbf{x}_i - \mathbf{b}^{(y_i)}) \odot \frac{1}{\mathbf{w}^{(y_i)}}$
5:     Append $(\mathbf{x}_i, y_i)$ to $D_{\widetilde{c}}$
6: **end for**
7: **Output** Recovered dataset $D_{\widetilde{c}}$.

---

**Optimization** We solve (3.1) by adopting projected gradient descent (Madry et al., 2018) to iteratively update the two parameter sets $\mathbf{W}$ and $\mathbf{B}$. Specifically, we first optimize over $\boldsymbol{\theta}$ for $I$ batches on the perturbed data via stochastic gradient descent. Then $\boldsymbol{\psi}$, namely $\mathbf{W}$ and $\mathbf{B}$, is updated through optimizing the inner minimization for $J$ steps. In particular, $\mathbf{W}$ and $\mathbf{B}$ are iteratively updated by:

$$\mathbf{w}_{t+1}^{(y_i)} = \mathbf{w}_t^{(y_i)} - \eta_1 \nabla_{\mathbf{w}} \mathcal{L}(f_\theta(\mathbf{w}_t^{(y_i)} \odot \mathbf{x}'_i + \mathbf{b}_t^{(y_i)}), y_i), \tag{3.3}$$

$$\boldsymbol{b}_{t+1}^{(y_i)} = \boldsymbol{b}_t^{(y_i)} - \eta_2 \nabla_{\boldsymbol{b}} \mathcal{L}(f_\theta(\mathbf{w}_t^{(y_i)} \odot \mathbf{x}'_i + \mathbf{b}_t^{(y_i)}), y_i), \tag{3.4}$$

at step $t + 1$ with learning rates $\eta_1$ and $\eta_2$. In practice, the iterations on $J$ will traverse one pass (epoch) through the whole dataset. This typically allows $\mathbf{W}$ and $\mathbf{B}$ to better capture the strong (false) correlation between the perturbations and the model. After each update, each entry of $\mathbf{W}$ is clipped into to the range $[1 - \epsilon/2, 1 + \epsilon/2]$, and each entry of $\mathbf{B}$ is constrained to $[-\epsilon/2, \epsilon/2]$. Moreover, $D_p$ is updated at each iteration according to (3.2). The iteration stops when $f_{\boldsymbol{\theta}}$ obtains a low training error, controlled by $\lambda$, on the perturbed dataset.

To unlock the dataset, we simply do an inverse of the linear transformation given the corresponding $\mathbf{W}$ and $\mathbf{B}$ from Algorithm 1. Contrary to the locking process, we first apply an inverse shifting and then apply an inverse scaling operation. Detailed process is presented in Algorithm 2. We note that the whole learnability control process based on a linear transformation is nearly lossless in terms of information[1].

---

[1] In practical image classification tasks, the perturbed images are further clipped into $[0, 1]$ range and thus may slight lose a bit information during the unlocking process. Yet the difference is neglectable. See more details in Appendix B.4.

### 3.1.2 CONVOLUTIONAL FUNCTIONAL PERTURBATION

Next, we present another way of applying convolutional function to craft adversarial perturbations. To begin with, we quickly review the idea of invertible ResNet (i-ResNet) (Behrmann et al., 2019).

**i-ResNet** Residual networks consist of residual transformations such as $\mathbf{y} = \mathbf{x} + h_\psi(\mathbf{x})$. Behrmann et al. (2019) note that this transformation is invertible as long as the Lipschitz-constant of $h_\psi(\cdot)$ satisfies $Lip(h_\psi) \leq 1$. Behrmann et al. (2019) proposed the invertible residual network that uses spectral normalization to limit the Lipschitz-constant of $h_\psi(\cdot)$ and obtain invertible ResNet. In particular, the inverse operation can be done by performing an fix-point iteration starting from $\mathbf{x}_0 = \mathbf{y}$, and then iteratively compute $\mathbf{x}_{t+1} = \mathbf{x}_t - h_\psi(\mathbf{x}_t)$. We refer the reader to the original paper for more detailed illustrations.

The invertible residual block is a perfect match for our purposes: we need an invertible and parameter-efficient transformation for data encryption/decryption, while the invertible residual block only requires a few convolutional layers (shared convolution filters). Specifically, let's denote the class-$y$'s invertible residual transformation function[2] as $g^{(y)}$. For each $\mathbf{x}_i$ with label $y_i \in \{1, ..., K\}$, we design the following the transformation function:

$$\mathbf{x}'_i = g^{(y_i)}(\mathbf{x}_i) = \mathbf{x}_i + \epsilon \cdot \tanh(h_\psi^{(y_i)}(\mathbf{x}_i)), \tag{3.5}$$

where each $h_\psi^{(y)}$ is composed of multiple spectrally normalized convolution layers with same-sized output, and ReLU activations, in order to make the transformation invertible. The output of $h_\psi^{(y)}(\cdot)$ can be viewed as the perturbation noise added to the original image. To limit the $L_\infty$ norm of the perturbation to satisfy the $\epsilon$ constraint, we add an extra tanh function outside $h_\psi^{(y)}(\cdot)$ and then multiplied by $\epsilon$. Note that this does not change the invertible nature of the whole transformation, as the tanh function is also 1-Lipschitz.

The crafting process is similar to the linear transformation case, as summarized in Algorithm 3 in appendix. Specifically, we adopt stochastic gradient descent for updating $\boldsymbol{\theta}$ and $\boldsymbol{\psi}$ alternatively. To unlock a perturbed data sample in this setting, we start from $\mathbf{x}_i = \mathbf{x}'_i$ and perform fixed point iterations by updating $\mathbf{x}_i = \mathbf{x}_i - \epsilon \cdot \tanh(h_\psi^{(y_i)}(\mathbf{x}_i))$, as shown in Algorithm 4 in the appendix. Note that although this iterative unlocking process cannot exactly recover the original dataset, as shown in Appendix B.4, the actual reconstruction loss is minimal.

## 4 EXPERIMENTS

In this section, we empirically verify the effectiveness of our methods for learnability control. Following Huang et al. (2021), we then examine the strength and robustness of the generated perturbations in multiple settings and against potential counter-measures. Tasks are evaluated on three publicly available datasets: CIFAR-10, CIFAR-100 (Krizhevsky et al., 2009), and IMAGENET (Deng et al., 2009). Specifically, for IMAGENET experiments, we picked a subset of 10 classes, each with 700 training and 300 testing samples, denoted by IMAGENET-10. Detailed experimental setups can be found at Appendix B.1.

### 4.1 LEARNABILITY CONTROL

Our main experiments aim to answer the following two key questions related to the learnability control task: (1) are the proposed transformations effective for controlling (locking and unlocking) the learnability of the dataset? (2) is this learnability control strategy universal (agnostic to network structures or datasets)? We test the strength of our methods on standard image classification tasks for CIFAR-10, CIFAR-100, and IMAGENET-10 datasets on four state-of-art CNN models: ResNet-18, ResNet-50 (He et al., 2016), VGG-11 (Simonyan & Zisserman, 2014), and DenseNet-121 (Huang et al., 2017). We train each model on CIFAR-10 for 60 epochs, CIFAR-100 for 100 epochs, and IMAGENET-10 for 60 epochs.

As shown in Table 1, both of the proposed adversarial transformations (linear or convolutional) can reliably lock and restore the dataset learnability. Specifically, compared to clean models trained on

---

[2]The transformation is designed to keep the input dimension unchanged.

$D_c$, models trained on perturbed dataset $D_p$ have significantly worse validation accuracy. This effect is equally powerful across network structures with different model depths. On the other hand, the learnability is ideally recovered through the authorization process as the model trained on recovered dataset $D_{\widetilde{c}}$ achieves comparable performance as the corresponding clean models. This is also true on IMAGENET-10 dataset, suggesting that the information loss through the learnability control process is negligible for training commercial models. We can also observe that the linear transformation function seems more powerful than the convolutional one in terms of the validation accuracy on the learnability locked dataset $D_p$. However, the convolutional lock is advantageous in that the structure of $h_\psi$ can be flexibly adjusted to control the number of parameters in the key[3], while the linear one has the key size strictly proportional to the data dimension. Also note that while we use ResNet-18 model for $f_\theta$ in the learnability locking process, other model architectures could also be effective for our task, which is further discussed in Appendix C.4.

Table 1: Test accuracy for models trained on clean dataset ($D_c$), perturbed dataset($D_p$), and recovered clean dataset ($D_{\widetilde{c}}$). The first row summarizes result for linear perturbation function; the second row lists the result for convolutional perturbation function.

| Transform | Network | CIFAR-10 | | | CIFAR-100 | | | IMAGENET-10 | | |
|---|---|---|---|---|---|---|---|---|---|---|
| | | $D_c$ | $D_p$ | $D_{\widetilde{c}}$ | $D_c$ | $D_p$ | $D_{\widetilde{c}}$ | $D_c$ | $D_p$ | $D_{\widetilde{c}}$ |
| Linear | ResNet-18 | 91.60 | 12.38 | 90.66 | 68.46 | 8.58 | 67.16 | 81.57 | 9.97 | 81.90 |
| | ResNet-50 | 94.28 | 15.62 | 95.59 | 70.12 | 6.98 | 69.09 | 84.20 | 10.30 | 83.53 |
| | VGG-11 | 90.76 | 15.07 | 91.37 | 67.25 | 6.62 | 67.49 | 81.07 | 11.80 | 81.33 |
| | DenseNet-121 | 94.36 | 14.75 | 93.33 | 70.81 | 7.48 | 71.32 | 83.17 | 8.53 | 83.37 |
| Conv | ResNet-18 | 91.02 | 16.60 | 91.95 | 69.27 | 8.92 | 68.93 | 82.83 | 12.20 | 81.70 |
| | ResNet-50 | 95.68 | 19.39 | 94.79 | 69.58 | 8.70 | 67.29 | 84.93 | 10.37 | 84.63 |
| | VGG-11 | 91.07 | 19.61 | 90.88 | 67.32 | 7.45 | 67.98 | 81.30 | 13.20 | 81.90 |
| | DenseNet-121 | 93.22 | 20.23 | 93.48 | 71.14 | 5.62 | 71.54 | 84.37 | 12.47 | 84.43 |

**Control over Single Class:** In practice, the data controller may not has access to the entire dataset, rather, the access may be restricted to single or several classes of the whole dataset. We test our learnability control process with only one single class being manipulated. Specifically, we choose to perturb the class "bird" in the CIFAR-10 dataset and observe the corresponding learnability performance. In Figure 2, we show the heatmap of the prediction confusion matrices on our single-class controlled CIFAR10 dataset via both linear and convolutional transformations. It can be observed that our proposed method reliably fools the network into misclassifying "bird" as other class labels. This demonstrates the effectiveness of our proposed method even with single class data being perturbed. On the other hand, the model trained on the unlocked "bird"-class data can achieve 93% and 91% test accuracies respectively for linear and convolutional perturbation functions. This confirms that our learnability lock can be flexibly adapted to different protection tasks. Due to the space limit, we refer the readers to Section B.2 in the appendix for experiments with learnability control on multiple classes.

**Effect of Perturbation Limit $\epsilon$:** We also test the effectiveness of our learnability control with different $\epsilon$ limitations. We conduct two sets of experiments on the CIFAR10 dataset using linear transformation with $\epsilon = 8$ and $\epsilon = 16$ respectively. As shown in Figure 3, the learning curves demonstrate that larger perturbation provides apparently better protection to the data learnability. While it is true that larger $\epsilon$ gives better learnability control on the dataset, we point out that the too large noise will make the perturbation visible to human eyes.

## 4.2 ROBUSTNESS ANALYSIS

In this section, we evaluate the robustness of the functional perturbations against proactive mitigation strategies. It is worth noting that there is no existing strategy specifically targeted to compromise the learnability control process. So we consider two types of general defenses that may affect the effectiveness of our learnability lock: 1) data augmentations or filtering strategies that preprocess the

---

[3]In many cases, the CNN structure with shared convolutional filters requires much fewer parameters. The detailed structure used for our convolutional transformation can be found in the Appendix.

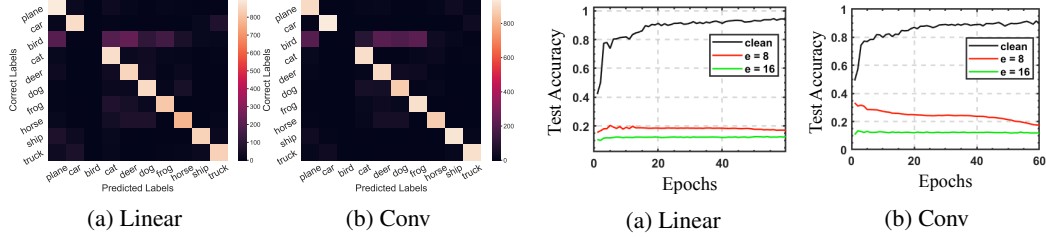

| (a) Linear | (b) Conv | (a) Linear | (b) Conv |

Figure 2: Heatmap of the classification confusion matrix for learnability control on the single class "bird" of CIFAR-10 for (a) linear transformation, and (b) convolutional transformation.

Figure 3: Test accuracy against training epochs for model training on the perturbed dataset $D_p$ using different $\epsilon$ for (a) linear transformation, and (b) convolutional transformation.

input data which may mitigate the effect of our perturbations; 2) adversarial training based defensive techniques that target to learn the robust features in the input which still exist in the perturbed data.

### 4.2.1 DATA AUGMENTATION AND FILTERING TECHNIQUES

It is possible that data augmentation can be used to mitigate the effect of small adversarial perturbations. In addition to standard augmentation methods such as rotation, shifting, and flipping, we test our method against several more advanced ones, including Mixup (Zhang et al., 2018), Cutmix (Yun et al., 2019), and Cutout (Cubuk et al., 2018). We also evaluate our method on some basic filtering techniques such as random $L_\infty$ noises and Gaussian smoothing restricting the perturbation strength $\epsilon = 8$. Experimental results in Table 2 indicate that none of these techniques defeat our learnability control method as the model trained on the augmented samples still behaves poorly. Standard augmentations such as rotation, shifting and flipping can only recover an extra $1\%$ of the validation accuracy. Mixup (Zhang et al., 2018) achieved the best learnability recovery result of $43.88\%$ for linear lock and $36.97\%$ for convolutional lock, yet it is still far from satisfactory for practical use cases. Note that although convolutional based perturbations achieve worse learnability locking performance compared with linear perturbations, they are actually more robust to such data augmentation or filtering techniques. It is also worth noting that in our experiment, we assume that the data provider doesn't know the potential augmentations used for generating the learnability locked dataset. We believe our method can be further improved by crafting the learnability lock on an augmented dataset so that the lock can generalize better to various augmentations.

Table 2: Summary of applying data augmentation and filtering techniques on the learnability locked dataset.

| Defenses | Acc (Linear) | Acc (Conv) |
|---|---|---|
| None | 14.79 | 17.61 |
| Random Noise | 19.83 | 17.32 |
| Gaussian Blurring | 15.59 | 21.28 |
| rotate & flip & crop | 15.25 | 18.81 |
| Cutmix | 20.72 | 18.60 |
| Cutout | 26.28 | 23.04 |
| Mixup | 43.88 | 36.97 |

Table 3: Summary on the effectiveness of applying adversarial training on several learnability attack methods and our proposed learnability lock.

| Method ↓ | Val Acc |
|---|---|
| Unlearnable Examples | 85.89 |
| Gradient Alignment | 83.56 |
| Adversarial Poisoning | 86.09 |
| **Learnability Lock (linear)** | 65.53 |
| **Learnability Lock (conv)** | 71.78 |

### 4.2.2 ADVERSARIAL TRAINING

Adversarial training (Madry et al., 2018) can also be viewed as a strong data augmentation technique against data perturbations. It is originally designed to help the model learn robust features to defend against the adversarial examples, which are generated by solving a min-max problem. Note that this min-max optimization is the exact counterpart of the min-min formulation in Huang et al. (2021). Huang et al. (2021) reported that adversarially trained models can still effectively learn the useful information from the dataset perturbed by the error-minimizing noise (gaining around $85\%$ validation accuracy). We test our methods by adversarially training a ResNet-50 model on the perturbed

CIFAR-10 dataset created from functional perturbations. We compare the result with three existing learnability attack methods including unlearnable examples (Huang et al., 2021), gradient alignment (Fowl et al., 2021b), and adversarial poisoning (Fowl et al., 2021a). The attack used for adversarial training is the traditional PGD approach with 10 iteration steps, and the maximum $L_\infty$ perturbation is limited to $8/255$.

Experimental results in Table 3 indicate that the learnability lock outperforms existing methods based on additive noises in terms of adversarially trained models. We suspect this is because simple transformations, such as scaling and shifting, are able to generate much more complicated perturbation patterns than pure additive noises (Laidlaw & Feizi, 2019). A similar finding is noted by Zhang et al. (2019b) that simple transformations can move images out of the manifold of the data samples and thus expose the adversarially trained models under threats of "blind spots". While adversarial training proves to be much more effective than other data augmentations techniques, it also fails to recover the validation accuracy back to a natural level and is, on the other hand, extremely computationally expensive. In Appendix B.3, we further discuss a hypothetical case where attacker realizes the transformation function used for learnability control and thus perform an adaptive adversarial training, which is shown to be even less effective than standard adversarial training.

### 4.3 PRACTICAL GUIDELINES

In this section we study some unique properties of our learnability control framework. Since our proposed learnability control framework allows us to restore its learnability with the correct secret key, one may wonder whether this learnability key is unique and reliable for practical use. In other words, if we generate two sets of keys for two clients based on one common data source, would the two keys mutually

Table 4: Summary of uniqueness experiment results with a ResNet-18 trained on CIFAR-10

| Keys ↓ | Linear (A) | Conv (A) |
|---|---|---|
| **Linear (B)** | 14.79 | 11.35 |
| **Conv (B)** | 19.83 | 17.81 |

unlock the other "learnability locked" dataset? This is crucial since if it is the case, then as long as one set of key (transformation function) is exposed to an attacker, we lost the learnability locks for all the perturbed datasets based on the same source data. In our experiment, we train two learnability locks for each transformation separately on the CIFAR-10 dataset and generate learnability locked datasets $D_p^1$ and $D_p^2$ with corresponding keys $\psi_1$ and $\psi_2$. Then we use $\psi_1$ to unlock $D_p^2$ and train a model on the retrieved dataset. As shown in Table 4, each row represents the transformation method we used as the key and each column stands for a learnability locked dataset we intend to unlock. The result suggests that learnability lock is strongly tied with a key that cannot be used to mutually unlock other locks, which introduces the desired property that prevents any unwanted data usage caused by partial leakage of secret keys.

We also conduct various ablation studies on the practicality of learnability lock including the effect of different perturbation percentages in $D_p$, different model architectures of $f_\theta$, the possibility of using global transformation functions or a mixture of transformation functions, for which we refer the reader to Appendix C.

## 5 CONCLUSION

We introduce a novel concept *Learnability Lock* that leverages adversarial transformations to achieve learnability control on specific dataset. One significant advantage of our method is that not only we can attack the learnability of a specific dataset with little visual compromises through an adversarial transformation, but also can easily restore its learnability with a special key, which is light-weighted and easy-to-transfer, through the inverse transformations. While we demonstrate the effectiveness of our method on visual classification tasks, it is possible to adapt our method to other data formats, which we leave as future work directions.

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

# A    Algorithms for Convolutional Learnability Locking and Unlocking

In this section, we attach the complete algorithms for the convolutional learnability locking and unlocking process. Specifically, the convolutional learnability locking process is summarized in Algorithm 3. The general procedure is similar to Algorithm 1: we update the perturbed dataset $D_p$ with the current convolutional function and then alternatively update $\boldsymbol{\theta}$ and $\boldsymbol{\psi}$ using the perturbed data via stochastic gradient descent.

The unlocking steps are summarized in Algorithm 4. Specifically, the inverse mapping of the residual network can be approximated by a fixed-point iteration. Note that while this fixed-point iteration may not perfectly reconstruct the original data, we show that this information loss is essentially negligible (in terms of both reconstruction error and learnability) in Section B.4.

---

**Algorithm 3** Learnability Locking (convolutional)

---

1: **Inputs:** Clean dataset $D_c = \{(\mathbf{x}_i, y_i)\}_{i=1}^n$, crafting model $f$ with parameters $\boldsymbol{\theta}$, transformation $h$ with randomly initialized parameters $\boldsymbol{\psi}_0$, perturbation bound $\epsilon$, outer optimization step $I$, inner optimization step $J$, error rate $\lambda$, loss function $\mathcal{L}$
2: **while** error rate $< \lambda$ **do**
3:     $D_p \leftarrow$ empty dataset
4:     **for** $(\mathbf{x}_i, y_i)$ **in** $D_c$ **do**
5:         $\mathbf{x}'_i = \mathbf{x}_i + \epsilon \cdot \tanh(h_{\boldsymbol{\psi}}^{(y_i)}(\mathbf{x}_i))$
6:         Add $(\mathbf{x}'_i, y_i)$ to $D_p$
7:     **end for**
8:     **repeat** $I$ steps:                                  ▷ outer optimization over $\theta$
9:         $(\mathbf{x}'_k, \mathbf{y}_k) \leftarrow$ Sample a new batch from $D_p$
10:         Optimize $\boldsymbol{\theta}$ over $\mathcal{L}(f_\theta(\mathbf{x}'_k + \epsilon \cdot \tanh(h_{\boldsymbol{\psi}}^{(\mathbf{y}_k)}(\mathbf{x}'_k))), \mathbf{y}_k)$ by SGD
11:     **repeat** $J$ steps:                                  ▷ inner optimization over $\psi$
12:         **for** $(\mathbf{x}'_i, y_i)$ **in** $D_p$ **do**
13:             Optimize $\boldsymbol{\psi}$ over $\mathcal{L}(f_\theta(\mathbf{x}'_i + \epsilon \cdot \tanh(h_{\boldsymbol{\psi}}^{(y_i)}(\mathbf{x}'_i))), y_i)$ by SGD
14:         **end for**
15:     error rate $\leftarrow$ Evaluate $f_{\boldsymbol{\theta}}$ over $D_p$
16: **end while**
17: **Output** Poisoned dataset $D_p$ , key $\boldsymbol{\psi}$

---

---

**Algorithm 4** Learnability Unlocking (convolutional)

---

1: **Inputs:** Poisoned dataset $D_p = \{(\mathbf{x}'_i, y_i)\}_{i=1}^n$, transform function (i-ResNet) $h_{\boldsymbol{\psi}}$, number of fixed-point iterations $m$, perturbation bound $\epsilon$
2: $D_{\widetilde{c}} \leftarrow$ empty list
3: **for** each $\mathbf{x}'_i, y_i$ in $D_p$ **do**
4:     **for** i = 1, ..., $m$ **do**
5:         $\mathbf{x}_i = \mathbf{x}'_i - \epsilon \cdot \tanh(h_{\boldsymbol{\psi}}^{(y_i)}(\mathbf{x}_i))$
6:     **end for**
7:     Append $(\mathbf{x}_i, y_i)$ to $D_{\widetilde{c}}$
8: **end for**
9: **Output** Recovered dataset $D_{\widetilde{c}}$ .

---

# B    Additional Experiment Details

## B.1    Experimental Setup

For all experiments, we use ResNet-18 (He et al., 2016) as the base model for generating the learnability lock unless specified. Performance of the learnability locking process is evaluated by the

Table 5: Structure of transformation model $h$ for convolutional learnability lock on CIFAR

| Layer Type | # channels | Filter Size | Stride | Padding | Activation |
|------------|------------|-------------|--------|---------|------------|
| Conv | 8 | $3 \times 3$ | 1 | 1 | ReLU |
| Conv | 16 | $3 \times 3$ | 1 | 1 | ReLU |
| Conv | 16 | $1 \times 1$ | 1 | 0 | ReLU |
| Conv | 8 | $3 \times 3$ | 1 | 1 | ReLU |
| Conv | 3 | $3 \times 3$ | 1 | 1 | ReLU |

testing accuracy of the model train from scratch on the controlled dataset $D_p$. Similarly, effectiveness of the unlocking process is examined by the testing accuracy of the model trained on the learnability restored dataset $D_{\widetilde{c}}$. To ensure the stealthiness of the perturbation, we restrict $\epsilon = 8/255$ for CIFAR-10 and CIFAR-100, and $\epsilon = 16/255$ for IMAGENET. To evaluate the effectiveness of the learnability control, the evaluation models are trained using Stochastic Gradient Descent (SGD) (LeCun et al., 1998) with initial learning rate of $0.01$, momentum of $0.9$, and a cosine annealing scheduler (Loshchilov & Hutter, 2016). For model training during the learnability locking process (updating $f_{\boldsymbol{\theta}}$), we set initial learning rate of SGD as $0.1$ with momentum as $0.9$ and cosine annealing scheduler without restart. Cross-entropy is always used as the loss function if not mentioned otherwise. The batch size is set as $256$ for CIFAR-10 and CIFAR-100, $128$ for the IMAGENET due to memory limitation.

**Setup for Linear Learnability Lock** For learnability locking using linear transformations, we use the following hyper-parameters in Algorithm 1: for CIFAR-10, we set $I = 20$ and $J = 1$ with a learning rate over $\psi$ of $0.1$ according to (3.3) and (3.4); for CIFAR-100, we let $I = 25$ and $J = 3$ with learning rate $0.1$; for IMAGENET-10 dataset, we set $I = 30$ and $J = 5$ with learning rate $0.1$. In addition, the functional parameters $\psi = \{\mathbf{W}, \mathbf{B}\}$ are initialized with all ones to $\mathbf{W}$ and all zeros to $\mathbf{B}$. For the exit condition, $\lambda$ is set to $0.1$.

**Setup for Convolutional Learnability Lock** The convolutional model $h_{\psi}$ we used for generating perturbations is demonstrated in Table 5. For experiment on IMAGENET, we use the same $h$ structure but modify the number of channels in order to 64, 128, 128, 64, 3. The hyper-parameters are set as follows in Algorithm 3: for CIFAR-10, we use $I = 25$ and $J = 3$ with a learning rate of $0.1$ with SGD over $\psi$; for CIFAR-100, we set $I = 30$ and $J = 5$ with a initial learning rate $0.05$, which shrinks by half for every 5 iterations of Algorithm 3. On IMAGENET, we instead use $I = 40$ and $J = 5$ with learning rate $0.01$. For the unlocking process, we set the number of fixed-point iterations $m = 5$. For exit condition, we set $\lambda = 0.2$ and also applied early stopping to avoid overfitting.

### B.2 LEARNABILITY CONTROL ON MULTIPLE CLASSES

We follow the experimental setup in Section 4.1 and consider the case where multiple, but not all, classes need to be protected. For linear transformation, we select 4 classes ("automobile", "cat", "dog", "horse") from the CIFAR-10 dataset to conduct learnability control. For convolutional transformation, we do the same on another 4 classes ("bird", "deer", "frog", "ship") for demonstration. Note that the controlled classes are selected by random and our method also generalizes to other settings. Figure 5 shows the prediction results of a ResNet-50 trained on the partially controlled dataset. We can observe that model falsely predict almost all the samples that belong to classes under learnability control, while remaining unaffected on other classes. The learnability can also be reliably restored by unlocking the controlled classes. Models trained on the restored datasets achieve testing accuracies of $94\%$ and $93\%$, respectively.

### B.3 MORE DETAILS ON ADVERSARIAL TRAINING

Generally speaking, adversarial training fails to serve as an ideal counter-measure to our proposed learnability control. Aside from the extra computational cost to craft adversarial examples in each training step, it has been shown that adversarially trained models suffer from a compromised natural accuracy (Zhang et al., 2019a; Wu et al., 2020). Our experiments indicate that with the same training pipeline, the adversarially trained model on a CIFAR-10 dataset can achieve at best $87\%$ validation accuracy, while the normally trained model can easily attain a $93\%$. The degradation is even more apparent on a industry-level dataset with high resolution (Fowl et al., 2021a). In addition,

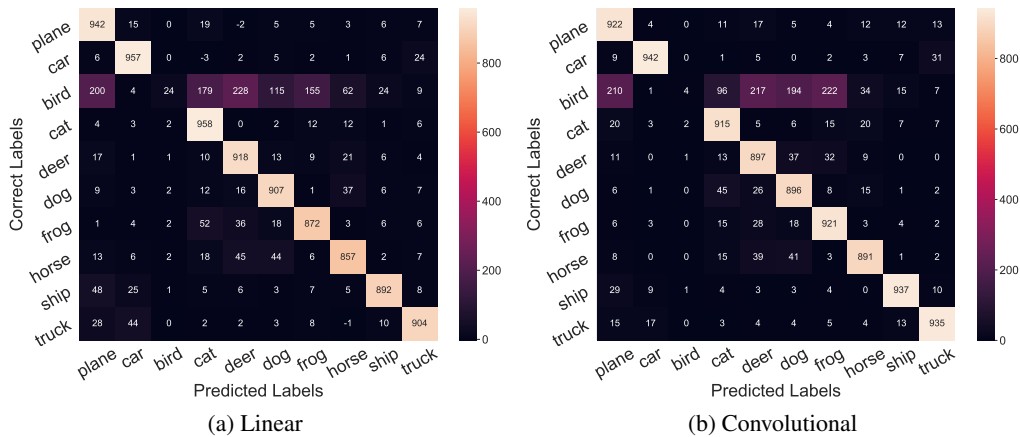

Figure 4: Annotated version of Fig. 2 – classification confusion matrix for learnability control on the single class "bird" of CIFAR-10 for (a) linear transformation, and (b) convolutional transformation.

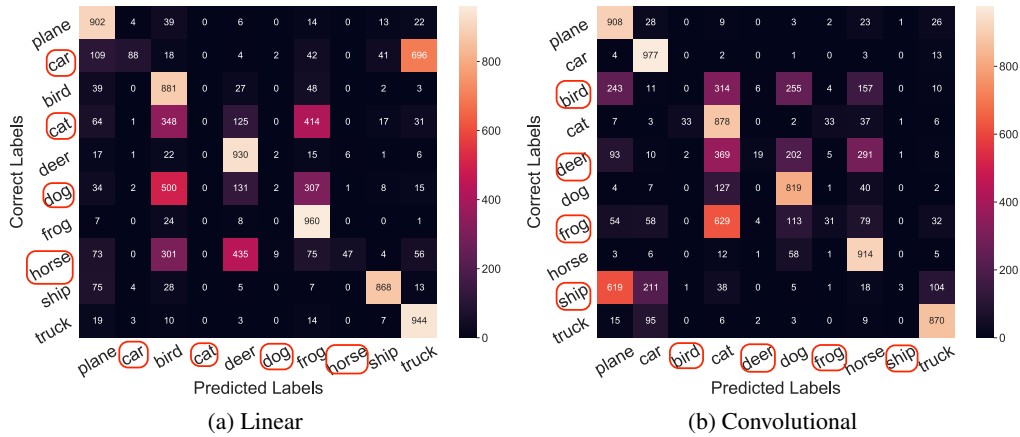

Figure 5: Prediction results of a ResNet-50 model trained on partially learnability-controlled CIFAR-10. The classes under control are circled in red. It is easy to see that the model has fairly bad performance on learnability-controlled classes, while nearly unaffected on other classes.

the adversarial training could be less effective without knowing the exact method an adversary used for the attack. This is also true in our learnability control setting as the adversary is unaware of the transformations used within the learnability lock.

To further explore the stability of our learnability control strategy, we envision a hypothetical scenario that the adversary actually know what type of transformation is used for our learnability lock. Therefore, the adversary may choose to adversarially train the model using the exact same type of transformations for solving the min-max problem. Specifically, if the learnability lock is crafted based on a linear transformation, then the adversary can also first solve the maximizer for $\mathbf{W}$ and $\mathbf{B}$ that mostly degrades the model performance at each training step and then solve the minimizer on the outer model training problem. We test out this concern for both the linear and convolutional learnability control processes. Specifically, we adversarially train a ResNet-50 on the CIFAR-10 dataset based on class-wise noises with linear and convolutional transformations respectively. In addition, we also use the sample-wise linear transformation (which is much computationally intensive) to craft the noise for adversarial training. We are not able to leverage the sample-wise convolutional transformation for the full adversarial training process due to the high computational burden. Instead, we craft the sample-wise noises based on convolutional transformation on a well-trained model on $D_p$, and include the perturbed samples as data augmentation to $D_p$. We then continue training the model for 20 iterations. The result is summarized in Figure 6, suggesting that none of the adversarial training techniques can reliably defeat our method. From Figure 6 we also observe

that when the adversary is unaware of the type of transformation used, the best strategy is probably to apply the standard adversarial training, even though its learnability recovery ability is also limited (as previously shown in Section 4.2.2).

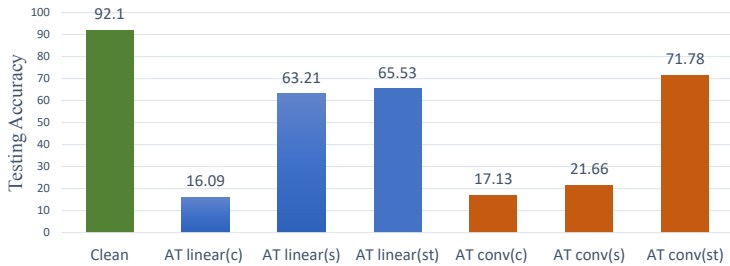

Figure 6: Performance on the learnability locked dataset trained by adapted adversarial training with respect to the transformation functions. Specifically, "c" stands for adversarial training with class-wise perturbations, "s" means sample-wise, and "st" means the standard adversarial training using PGD.

### B.4 RECONSTRUCTION LOSS

Normally the encryption and decryption process is accompanied by a certain degree of information loss. This is also true for the linear and convolutional learnability lock. On the one hand, the samples transformed by the lock may be further clipped into range $[0, 1]$ in image classification tasks. This introduces slight information loss when we reconstruct the samples with inverse transformation. Additionally, for the convolutional lock, the inverse transformation is approximated by fixed-point iterations, bringing a certain degree of information loss. While we already show in Table 1 that this loss is negligible for restoring the learnability, here we explore the ability to reconstruct the original dataset with the two proposed transformations. The experiment is conducted on CIFAR-10 with two pre-trained learnability locks. The information loss is measured by the average $L_2$ loss between retrieved images from $D_{\widetilde{c}}$ and original images from $D_c$, calculated on 500 randomly selected images. Results are included in Table 6. It shows that the information loss is trivial compared with the distance from perturbed images, and thus the learnability control process is rarely affected.

Table 6: Reconstruction losses for learnability locks with both linear and convolutional transformations measured in $L_2$ distance.

| Average $\mathbf{L_2}$ loss compared to $\mathbf{D_c}$ | $\mathbf{D_p}$ | $\mathbf{D_{\widetilde{c}}}$ |
|---|---|---|
| Linear | $0.81 \pm 0.07$ | $2.58\mathrm{e}{-7} \pm 9.78\mathrm{e}{-8}$ |
| Conv | $1.67 \pm 0.06$ | $4.91\mathrm{e}{-4} \pm 1.21\mathrm{e}{-3}$ |

## C ABLATION STUDY

In this section, we present some important ablation studies to our proposed learnability lock framework.

### C.1 PERTURBATION PERCENTAGES

In case when the entire training data collection is not available for the data controller to leverage learnability control, we examine the potency of our method when we perturb merely a portion of the training data. This is different from the aforementioned single-class experiments, which constrain the perturbation to only one single class. In this section, we study the effect if only part of the randomly selected samples from a full dataset has been controlled for learnability. Figure 7 shows that the effectiveness of both transformation functions is compromised when the training samples are not all under learnability control. The result suggests that the data provider should possess at

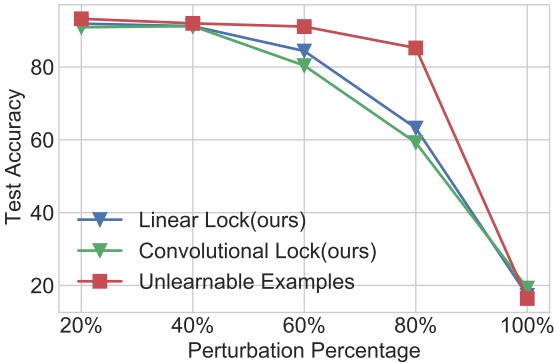

Figure 7: Test accuracy against perturbation percentage for models trained on learnability controlled dataset with different percentage of perturbed samples.

least 60% of the training data in order to activate the learnability control effect. This vulnerability is also noted in previous work (Huang et al., 2021). Compared with Huang et al. (2021)[4], we observed that both our linear and convolutional transformations achieve much better learnability locking performances especially when the perturbation percentage lies in the middle range (40% ∼ 80%).

## C.2 GLOBAL TRANSFORMATION FUNCTION

Our method focuses on class-wise transformation functions. Of particular interest is whether one global transformation function can be applied over the whole dataset and still lock the learnability. One advantage of applying a universal transformation is a smaller amount of parameters (size of the keys) needed. This makes transferring the keys between the data owner and clients more convenient. We test with both proposed perturbation functions on the CIFAR-10 dataset. Specifically, for the convolutional transformation, we use a deep network with 20 convolution layers (to ensure enough expressiveness for perturbing the entire dataset). As shown in Figure 8, while the potency of our method is decreased, the learnability lock is still effective in degrading the model performance by 20-30 percent. We believe that the performance can be further improved by either designing more effective perturbation functions or advanced optimization steps. We leave it for future study.

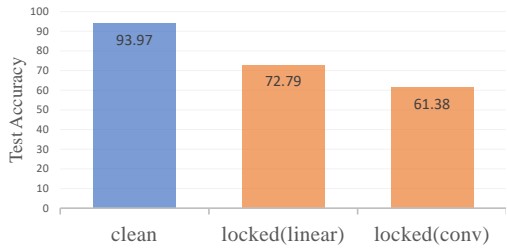

Figure 8: Experimental result for global transformation functions on CIFAR-10 dataset.

## C.3 MIXTURE OF PERTURBATION FUNCTIONS

Another way to improve the complexity of learnability control process against potential counter-measures is to apply a mixture of perturbation functions. In this sense, we can apply different transformation functions to each class in a random order so that the order itself could serve as another secret key. We show that this is possible by applying the linear transformation (Algorithm 1) on classes with odd labels and convolutional transformation (Algorithm 3) on classes with even labels from CIFAR-10. The result is listed in Table 7, generated on a ResNet-50 model. It is

---

[4]We directly use the data provided in the original paper of Huang et al. (2021) as the baseline.

Table 7: Performance comparison when using a mixture of perturbation functions.

| Perturbation ↓ | Val Acc |
|---|---|
| None | 93.89 |
| Linear | 13.93 |
| Conv | 17.06 |
| Mixture | 21.29 |

observed that using a combination of different locks has little impact on the potency of learnability control. Yet without knowing the exact transformation functions and the orders applied, it is harder for attackers to mitigate or reverse-engineer the perturbation patterns in order to compromise the protection.

### C.4 EFFECT ON CRAFTING MODEL ARCHITECTURES

While ResNet-18 is used as the default model architecture for crafting the learnability locked dataset in our experiments, we verify that other model architectures could serve the same purpose equivalently. Specifically, we use VGG-11 and DenseNet-121 as crafting models and evaluate the performance of the trained adversarial transformations on CIFAR-10 with ResNet-50. The results are listed in Table 8. For both linear and convolutional transformations, the learnability control process is still effective under different model architectures, suggesting that the learnability lock has the desired property that it is agnostic to network structures. In addition, we observe that a more powerful crafting model always generates a stronger learnability lock. For example, the validation accuracy of model trained on CIFAR-10 controlled by a lock crafted with DenseNet-121 is always lower than that crafted with VGG-11.

Table 8: Learnability locking performances with different model architectures of $f_\theta$.

| $g_\psi$ | $f_\theta$ | Test Acc (locked) | Test Acc (unlocked) |
|---|---|---|---|
| Linear | **VGG-11** | 17.53 | 91.35 |
| | **Densenet-121** | 12.01 | 90.34 |
| Conv | **VGG-11** | 19.21 | 91.67 |
| | **Densenet-121** | 14.56 | 92.75 |

### C.5 COMPARISON TO ADDITIVE PERTURBATION

Additive noise is widely adopted in the literature of adversarial machine learning. It usually has the form $\mathbf{x}' = \mathbf{x} + \boldsymbol{\delta}$ where $\boldsymbol{\delta}$ stands for the noise pattern being mapped onto a clean sample. Several recent works leverage additive perturbation to prevent models from learning useful features on a poisoned dataset (Fowl et al., 2021a; Huang et al., 2021; Fowl et al., 2021b). Existing additive noises can be summarized into two categories: sample-wise and class-wise. However, we argue that both kinds of them are not suitable for a task of learnability control due to several major drawbacks.

Sample-wise perturbation falls short of recovering the learnability for the whole dataset because of the extremely high cost to pass all the noise patterns to an authorized user. Obviously, the potential cost of transferring the "secret key" grows with the number of samples contained in the dataset, which tends to be large in a practical commercial setting. On the other hand, additive perturbations injected class-wise has much lower transferring cost. In specific, the secret key would involve a unique noise pattern for each class so that the cost is merely $O(d \times k)$, where $d$ stands for the sample dimension and $k$ represents the number of classes. However, the downside of class-wise additive perturbation is that it applies the same perturbation for all data in one class, and thus can be easily detected or reverse-engineered.

A transformation-based perturbation resolves both concerns raised above. First, the inversion cost is much lower than that of sample-wise additive noises. The proposed linear learnability lock involves

$O(d \times k)$ parameters to do the inverse transformation. This is merely $61440$ parameter values in the case of CIFAR-10. The cost Sof convolutional learnability lock is also low, with around $30000$ parameters on CIFAR-10 (based on $h$ provided in Table 5). On the other hand, the transformation based perturbation, though performed class-wise, can generate different noise patterns for each sample. This improves the stealthiness and stability of the learnability control process. Furthermore, it is shown that the transformation-based perturbation is more robust to standard adversarial training as counter-measure. Table 9 compares the maximum achievable accuracy of our method with Huang et al. (2021) that is based on additive noise. It shows that while the validation accuracy on the unlearnable data produced by our method is comparable to that of Huang et al. (2021), our method can ensure a much lower maximum achievable accuracy taking into account adversarial training. Table 10 summarizes the comparisons between our method and additive noises in terms of several significant properties for learnability control. Specifically, the "stealthiness" is reflected through diversity of functional perturbations rather than the value[5] of $\epsilon$. The diversity of functionally perturbations make it harder to reverse-engineer the noise patterns from the unlearnable dataset and thus break our protection. For example, suppose a key-lock pair (one noisy image and the original clean sample) is disclosed, then for the additive noise, it is easy for the attacker to detect or reverse-engineer the noise pattern and remove all of them. This is not true for our method as the functional parameters (e.g. an invertible resnet) are essentially impossible to recover with one key-lock pair.

Table 9: Comparison of maximum achievable accuracy with unlearnable examples (Huang et al., 2021). Here we focus on the **maximum** achievable validation accuracy of a ResNet-50 trained on the CIFAR-10 dataset using any training schemes.

|  | Normal Training | Adv Training | Max Achievable Accuracy |
|---|---|---|---|
| Unlearnable | 13.45 | 85.89 | 85.89 |
| Ours (linear) | 15.62 | 65.53 | **65.53** |
| Ours (conv) | 19.39 | 71.78 | **71.78** |

Table 10: Comparison of different strategies under the setting of learnability control. The "stealthiness" evaluates based on if the noise is easily detectable or can be reverse-engineered. (✓ denotes "yes", ✗ denotes "no").

| Perturbation ↓ | Inverse | Transfer Cost | Adv. Train | Stealthiness |
|---|---|---|---|---|
| Additive (sample-wise) | ✗ | high | ✗ | ✓ |
| Additive (class-wise) | ✓ | low | ✗ | ✗ |
| **Linear (ours)** | ✓ | low | ✓ | ✓ |
| **Convolutional (ours)** | ✓ | low | ✓ | ✓ |

---

[5]All the experiments apply the same $\epsilon$ for all baselines.

