# OpenReview forum: "Learnability Lock: Authorized Learnability Control Through Adversarial Invertible Transformations"
_ICLR.cc/2022/Conference — ICLR 2022 Poster_

### Official Review · Reviewer_nYcf · 2021-10-30

**Correctness:** 4
**Technical Novelty And Significance:** 3
**Empirical Novelty And Significance:** 3
**Recommendation:** 8
**Confidence:** 4

**Details Of Ethics Concerns:**

No ethical concerns are identified for this paper.

**Main Review:**

Strong points:
1. An advanced exploration of an important research direction. The idea of “learnability lock” appears fairly interesting, although it is motivated by the “unlearnable examples” idea. The “learnability lock” concept is one step closer to what these works try to achieve: giving data owners some real control of their own data without damaging (much) the original content.

2. The proposed invertible noise generation method looks simple and effective, should be easily applied in real-world scenarios.

3. The proposed method holds two advantages (invertible/controllability and robustness to adv train) over the prior work and these two advantages seem to be quite important for real-world usage.

4. The paper is well written, pleasant to read. The experiments are comprehensive.

Weak points:
1. It is not very clear why invertible transformation improves robustness to adversarial training. Although it explains somewhere, conceptually, multiplicative noise is more effective than additive noise, I think some experiments can be conducted to dig a bit deeper on this point. Looking forward to some explanations from the authors.

2. The robustness to adaptive adversarial training part in the appendix is also important. I suggest the authors provide a paragraph of analysis in the main text as well. Also, some experiments in the appendix are not mentioned in the main text.

3. The experiment structure looks quite similar to Huang et al. 2021, I suggest the author acknowledge their work/setting in the experimental setting.

4. The 100% unlearnable rate seems to be a remaining limitation of your and Huang’s work. Any thoughts on this?


**Summary Of The Paper:**

This paper introduces the idea of “learnability lock” for data authorization. This paper along with three prior works mentioned in the paper are all creative works recently proposed for data protection, an important research topic that has been investigated much before.  Compared to the “unlearnable examples” work of Huang et al. 2021, this work proposes to use inevitable linear/convolution transformations to formulate the noise. This was demonstrated to have two benefits that seem to address two important limitations of previous works: 1) making the unlearnable noise more controllable so “learnability” can be flexibly locked and unlocked; 2) the noise seem can survive (to some extent) adversarial training now.

**Summary Of The Review:**

Important research on data protection/authorization, simple yet very effective method, some level of technical improvements over prior work but addresses two key limitations, rich experiment, good writing. A quite solid paper overall.

---

> ### Author Response · Authors · 2021-11-18
> **Response to Reviewer nYcf**
>
> Thank you for your recognition and comments.
>
>
> *Q1:* Why invertible transformation improves robustness to adversarial training?
>
> *A1* We believe that the diversity and complexity of invertible transformations make our perturbations more powerful in tricking the model into believing the current parameters are perfect. Note that direct analysis towards this problem can be hard as our perturbation generation is a bi-level optimization and later adversarial training is another bi-level optimization, which makes the overall procedure fairly complicated. Luckily, robust models are generally believed to capture robust features which do not change by small perturbations. To empirically verify our conjecture, we evaluated unlearnable samples generated on a robust WideResNet-34 model pretrained using TRADES [1]. The result is shown below:
>
>
> | Noise | Accuracy | Sum of sample losses |
> | --- | --- | --- |
> |clean training set | 91.97 | 610.39|
> |Huang et al. 2021 | 92.03 | 609.79 |
> | ours (linear) |92.12 | 151.67|
> | ours (conv) | 92.21 | 149.19 |
>
> As shown from the table, the loss on datasets generated from both of our methods are much lower than that from Huang et al.’s unlearnable samples, which is quite similar to that of clean samples. This suggests that our generated perturbation noise can still trick the robust model to have lower loss values while Huang et al.’s perturbation noise has essentially no tricking effect towards the robust model.  In other words, the perturbations produced by our method are more relevant to robust features captured by adversarially trained models although the perturbation limit $\epsilon$ is the same. This partially accounts for why our method works better under adversarial training.
>
> ---
>
> *Q2:* I suggest the authors provide a paragraph of analysis on adaptive adversarial training.
>
> *A2:* Thanks for the suggestion! We have revised the experiment section accordingly to mention adaptive adversarial training experiments in the context and properly refer to all the experiments in the appendix.
>
> ---
>
> *Q3:* Acknowledgement of the experimental settings.
>
> *A3:* Thanks! We have revised the experiment section to acknowledge the experimental setting of Huang et al. 2021.
>
> ---
>
>  *Q4* Limitation on the unlearnable rate.
>
> *A4:* Although we have already improved the requirement of unlearnable rate compared with (Huang et al. 2021) as shown in Figure 6, we agree that both the current work and (Huang et al. 2021) requires a high unlearnable rate to ensure the effectiveness. We believe that it is an important future research direction. In fact, relaxing the limitation on the unlearnable rate is quite difficult, as a small amount of clean samples can help the model generalize useful features for the prediction tasks (see Figure 2(a)(b) in Huang et al. 2021).
>
> One plausible solution to relax the unlearnable rate is through changing the ordering for data supplied to the model. As shown in [2], by simply switching the order data points are fed to the model during training, the model performance can be significantly changed. In particular, they show that without any direct modification to the dataset, reshuffling the training data can be leveraged to approximate the SGD gradient update direction on an adversarial dataset, and thus effectively degrade the generalizability of the model. Following this idea, by manipulating the supply of unlearnable samples (i.e. the distribution of unlearnable samples in the dataset) during training process, it is possible to achieve unlearnability with a much lower unlearnable rate (with the access to manipulate the data order).
>
>
> [1] Zhang, Hongyang, et al. "Theoretically principled trade-off between robustness and accuracy." International Conference on Machine Learning. PMLR, 2019.
>
> [2] Shumailov, Ilia, et al. "Manipulating SGD with data ordering attacks." arXiv preprint arXiv:2104.09667 (2021).
>
> Thanks again and further feedback is more than welcomed.

---

> > ### Comment · Reviewer_nYcf · 2021-11-22
> > **Thanks for the clarification and new experiment**
> >
> > Thanks for the clarifications. I think most of my concerns are properly addressed. The unlearnable rate indeed seems to be a big challenge at the moment. And if a few unlearnable can break the entire dataset, then it will be a more powerful data poisoning attack and a big threat to deep learning in general. From the data protection, authorization, or IP protection perspective, I believe the authors have done quite good work compared to Huang et al. (2021). Learnability lock appears like the type of techniques people need to protect/authorize their social media data. I believe this work should be of significant interest to the community.
> >
> > I am happy to increase my rating to 8 (apparently, there is no 7 :)).

---

> > > ### Author Response · Authors · 2021-11-22
> > > **Thank you**
> > >
> > > We are glad that your concerns have been addressed. Thank you for your positive feedback and for increasing the score.

---

> ### Author Response · Authors · 2021-11-21
> **A friendly reminder of rebuttal conclusion**
>
> Thank you for your in-depth comments and suggestions to strengthen our manuscript. We hope that our responses could be helpful, and please let us know if you have any other questions or need more clarification.

---

### Official Review · Reviewer_ZETA · 2021-11-01

**Correctness:** 4
**Technical Novelty And Significance:** 3
**Empirical Novelty And Significance:** 3
**Recommendation:** 8
**Confidence:** 4

**Main Review:**

The idea of using invertible adversarial transformation for securing the data is interesting and novel. The paper is very well written and structured. Numerous experiments are conducted to show effectiveness of the method for breaking the ML models when "secured" data is used and for training accurate ML models when the inverse transformation is applied. The authors also show that their method is more robust to adversarial learning than state-of-the-art additive perturbation methods. The method is shown to be robust to strong data augmentation and filtering techniques which are often used as defense methods. Finally, the authors conduct experiment to show that when two different adversarial transformations are computed, their keys can not be used to unlock each other's transformation.

I have two questions for the authors:
1. What data was used to train the attacker model (f_theta)? Also, I can see that the ResNet-18 model was used for computing the attack, did you try to use other backbones? Does the attack still transfer to other model in this case? It would be great to elaborate on that somewhere in Experiments section.
2. In section 4.3 it is said "we train two learnability locks for each transformation separately". When learning two transformations from the same data, how can one control that the transformations will be different and that the model will not converge to the same solution twice?


**Summary Of The Paper:**

The paper proposes a new concept called "learnability lock" for securing distribution of sensitive data. The idea is to apply an invertible adversarial transformation when releasing the data, which makes the data "unlearnable" for machine learning models, but also preserves the visual properties of the data. Therefore, unauthorized practitioners can not use the data for training ML models, however authorized users can use specific key to invert adversarial perturbation and make the data "learnable" again.

The authors propose two invertible transformations to craft adversarial perturbations: linear pixel-wise transformation and convolutional functional transformation based on invertible ResNet. Numerous experiments demonstrate the effectiveness of the proposed transformations in both securing the data (making the data unlearnable when transformation is applied) and unlocking the transformation (making the data learnable when the transformation is inverted).

**Summary Of The Review:**

Overall, I believe that the idea of the paper is interesting and novel, the results are appealing. The writing is very clear and structured, the method is compared to other attacks and defenses and is shown to outperform previous methods. I vote for accepting the paper.

---

> ### Author Response · Authors · 2021-11-18
> **Response to Reviewer ZETA**
>
> Thank you for your positive comments on our paper.
>
>
> *Q1:* Data to train $f_{\theta}$:
>
> *A1:* The attacker model $f_{\theta}$ is trained on $D_p$ (the perturbed dataset). $D_p$ is initialized as the original clean dataset $D_c$. We then iteratively update $D_p$ during the training process as we update the parameters of the transformation function $g_{\phi}$.
>
> ---
> *Q2:* Different backbones and transferability to other models
>
> *A2:* While we use ResNet-18 as the crafting model for most experiments, we also tried other architectures as backbones. This has been discussed in Appendix C.4 in detail. Specifically, we use VGG-11 and Densenet-121 as crafting models and evaluate the performance of learnability locks generated respectively. As shown in Table 8, the learnability control process with both crafting models is still powerful when we use ResNet-18 as the learning model. This also suggests that the attack transfers across model structures regardless of the crafting model architecture. In addition, we observe that a more powerful crafting model usually generates a learnability lock that is more effective.
>
> ---
>
> *Q3:* How can one control that the transformations will be different and that the model will not converge to the same solution twice?
>
> *A3:* In fact, we do not need to explicitly control the transformations to be different. The training process will converge to different solutions due to different initializations of $\theta$. Empirically, the two sets of transformation functions are quite different just by using different initializations of $\theta$. To illustrate, we randomly take a set of 100 samples, denoted $X$, from the CIFAR-10 dataset and perturb them using two separately trained linear learnability locks. The resulting perturbed sample sets are denoted $X’_{1}$ and $X’_{2}$. We then compute the average l2 difference between each sample of the two perturbed sets and get the result as 30.71. For comparison, we also calculate the distance between the perturbed examples (i.e. $X’_{1}$ and $X’_{2}$) and the original clean samples $X$ and get the difference 41.03 and 45.04 respectively. This shows that the noise produced by two learnability locks are quite different from each other on concrete samples.
>
> Thanks again and further feedback is more than welcomed.

---

> > ### Comment · Reviewer_ZETA · 2021-11-26
> > **Thank you for the clarifications**
> >
> > I would like to thank the Authors for their clarifications, all my questions were addressed.

---

> ### Author Response · Authors · 2021-11-21
> **A friendly reminder of rebuttal conclusion**
>
> We sincerely appreciate your recognition and suggestions to strengthen our paper. Please let us know if you have further questions and we are pleased to address them before the rebuttal phase concludes.

---

### Official Review · Reviewer_BPLq · 2021-11-01

**Correctness:** 4
**Technical Novelty And Significance:** 2
**Empirical Novelty And Significance:** 2
**Recommendation:** 6
**Confidence:** 3

**Main Review:**

**Strengths**

1\. Well-motivated problem and reasonable results
- Given the prevalence of data being shared and made accessible everyday, it becomes crucial certain authorized uses e.g., preventing malicious entities on training a facial classifier on provided data. This paper takes an interesting step towards addressing this issue. The authors also show certain benefits of the approach e.g., controlling access to who can train models on the given data.

2\. Writing
- I highly appreciate the clarity in writing. The paper was clear for the most part and easy to follow.

**Major Concerns**

1\. Significance of contributions over prior work
- My first concern is the significance of the contributions over prior work and most notably with [1]. Similar to proposed approach, [1] also propose a bi-level minimization approach to introduce class-specific additive perturbations to prevent training on the perturbed dataset. My sub-concerns:
- (a) it seems that the improvement of the proposed approach is primarily addressing a specific failure mode -- being more robust to adversarial training on the perturbed dataset. Apart from that, [1] seems to outperform the proposed approach in few other scenarios e.g., when comparing test accuracy (Table 1) [1] appears to outperform (esp. on CIFAR-100 and ImageNet).
- (b) As for the technical contributions itself, the proposed approach also shares a bi-level minimization problem similar to [1]. But instead of learning fixed additive per-class perturbations, a network $g_\psi$ is trained to perturb the inputs. Could the authors comment on technical benefits over previous approach? I see that in Table 9 remarks the benefits as "adv. train" (which is supported) and "stealthiness" (which is unclear; aren't both methods evaluated at a comparable $\epsilon$?)

[1] "Unlearnable Examples: Making Personal Data Unexploitable" Huang et al., ICLR '21

2\. Optimization formulation
- I was also a bit unclear (or rather could not intuit) on how the optimization objective (3.1) makes the examples unlearnable.
- Specifically, wouldn't the optimal $g_\psi$ be an identity function which would lead to original 'clean' accuracy? In other words, if one were to take a pretrained $\theta$, I reckon $\psi$ would correspond to identity function?
- I'm also wondering how the approach learns good parameters $\psi$ in spite of an explicit term to maximize the loss of training on the perturbed dataset?
- I would appreciate if the authors slightly elaborated (to complement an already nice discussion below eq 3.1) on the optimization problem.

**Minor Concerns**

1\. Table 1
- I think the results in Table 1 is comparable to previous approaches (e.g., Table 1 of [1])? I would appreciate the authors also added corresponding rows (or alternatively a more descriptive table in appendix) so that direct comparison with prior work when possible. As a reader, I had to flip back and forth to compare the numbers.

2\. Unauthorized use of perturbed dataset for other tasks
- Are the authors aware whether training on $D_p$ fails only the envisioned task (e.g., 10-way multiclass classification over predefined CIFAR classes)? It seems problematic if the same images could be used for other tasks (e.g., face recognition) unforeseen by the creator.

**Nitpicks**
* "Control over Single Class": What is the accuracy of the model on locked bird class? What is the number in the heatmap of Fig. 2?

**Summary Of The Paper:**

- The paper tackles the problem of 'unlearnable examples': to perturb the images of a labeled dataset $D_c$ to obtain $D_p$ with the desiderata (a) training models on $D_p$ leads to models with significantly lower performance (b) image perturbations are constrained to some $\epsilon$-ball and (c) with the correct 'secret key' (learnable parameters in this case), one should approximate recover $D_c$.
- The approach `learnability lock' to make the examples unlearnable involve a bi-level minimization objective over the original network parameters $\theta$ and parameters $\psi$ of a 'secret key' network $g_\psi$ (which could be a linear layer or an invertible ResNet).
- Experiments demonstrate among other things: (i) training networks on $D_p$ leads to poor performances and on unlocked $D_c$ to high performances (ii) the proposed approach is more robust to augmentation strategies (specifically adversarial training) compared to previous approaches.

**Summary Of The Review:**

- Overall, the paper tackles a reasonably well-motivated idea of introducing perturbations to prevent unauthorized training over a labeled dataset. The authors show that the approach achieves this objective and is additionally more robust when compared to previous works.
- My biggest concern is the significance of improvements over prior work, which appears limited to a highly specific failure mode. I would be glad to increasing my score if the authors could address this.

---

> ### Author Response · Authors · 2021-11-18
> **Response to Reviewer BPLq  (Part 1/2)**
>
> Thank you for your detailed comments. We respond to your questions in order:
>
> *Q1:* Significance of contributions compared with (Huang et al. 2021).
>
> *A1:* While our method is built on the “error-minimizing noise” proposed by (Huang et al. 2021), we are considering a quite different scenario which can be thought of as data authorization. In short, Huang et al. propose to transform clean samples into unlearnable counterparts so that the dataset can be safely distributed. We, in addition, intend to transform those unlearnable counterparts back to clean samples so that authorized data users are allowed to train their models on the dataset (i.e. restore the learnability). This is more challenging than what Huang et al. resolve because the authorization process involves both a forward and backward transformation being the inverse of each other.
>
> ---
> *Q2:* Technical benefits of functional perturbations over previous approach
>
> *A2:* The functional perturbation could be advantageous from multiple perspectives: (1) compared to the additive perturbation used in (Huang et al. 2021), the transformation of functional perturbation is by design invertible so that authorized users can easily restore the “learnability” of the dataset by calling the inverse function; (2) the “key” used to unlock the dataset is much more light-weight than applying additive noise sample-wise; and the noise patterns generated by functional perturbations are more diverse for samples in the same class (while class-wise additive noise is universal); (3) the functional perturbation is shown to be more robust to adversarial training.
>
> In fact, the “stealthiness” is reflected through diversity rather than the value of $\epsilon$ (we use the same $\epsilon$ for all methods). The diversity of functionally perturbations make it harder to reverse-engineer the noise patterns from the unlearnable dataset and thus break our protection. For example, suppose a key-lock pair (one noisy image and the original clean sample) is disclosed, then for the additive noise, it is easy for the attacker to detect or reverse-engineer the noise pattern and remove all of them. This is not true for our method as the functional parameters (e.g. an invertible resnet) are essentially impossible to recover with one key-lock pair.
>
> In summary, our method adapts to an important application--data authorization--in the real world, and resolves three key limitations of previous works (based on additive noises) in this scenario.
>
> ---
> *Q3:* Intuition behind why optimization objective (3.1) can make examples unlearnable.
>
> *A3:* The objective is formulated as a bi-level optimization problem. The intuition is similar to the work of (Huang et al. 2021). Intuitively, the inner minimization will generate a perturbation such that the perturbed data is well fit the current model. Thus in the outer minimization task, the model is tricked into believing that the current parameters are perfect and there is little left to be learned. Therefore, such perturbations can make the model fail in learning useful features from data. Finally, training on such perturbed dataset will lead to a model that heavily relies on the perturbation noise. And when testing the model on clean samples, it cannot judge based on the noise pattern and thus makes wrong predictions. We have added more clarifications below equation 3.1 and hope this addresses your question.
>
> ---
> *Q4:* Optimal $g_{\psi}$ be an identity function on pretrained $\theta$
>
> *A4:* As we explained in A3, the inner minimization task aims to generate the perturbation such that the perturbed data is well fit the current model. Therefore, if we start from a pretrained $\theta$ that perfectly fits all training data, then the identity function might be optimal. However, since we are training $\theta$ from scratch, the training data does not fit the current model and thus the optimal $g_{\psi}$ we obtain is different from the identity function.
>
> ---
> *Q5:* Explicit maximization for unlearnable dataset
>
> *A5:* It is discussed in Section 4.1 of (Huang et al. 2021), the error-maximizing noise is much less effective than error-minimizing noise in terms of generating unlearnable examples. Therefore, we only consider the min-min formulation as in (Huang et al. 2021).

---

> > ### Comment · Reviewer_BPLq · 2021-12-07
> > **Thanks for the response**
> >
> > Thanks for the response, this does address some of my concerns.

---

> ### Author Response · Authors · 2021-11-18
> **Response to Reviewer BPLq  (Part 2/2)**
>
> *Q6:* Questions about Table 1
>
> *A6:* Thanks for the suggestion! We have added Table 9 (also attached below) in the Appendix to compare directly with Huang et al. 2021 on the unlearnable performances. Here we want to emphasize two things: (1) our method focuses on adding the “invertibility” apart from maintaining the “unlearnability” (See A1 and A2). Simply pursuing the ultimate “unlearnability” is not quite reasonable as a 10% accurate classifier is useless just like a 5% accurate classifier (no one would use them in practice); (2) when considering the “unlearnability”, we should consider the maximum achievable accuracy by any training schemes on one dataset. In this sense, we draw the following table and it shows that while the validation accuracy on the unlearnable data produced by our method is comparable to that of (Huang et al. 2021), our method can ensure a much lower maximum achievable accuracy taking into account adversarial training.
>
> |            | Normal Training | Adv. Training  | Maximum Achievable Accu.|
> | --------  | ----------- | -------- | ------|
> | Huang et al. 2021 | 13.45     |  85.89  | 85.89  |
> | Ours (linear)    | 15.62    |   65.53  | **65.53** |
> | Ours (conv)    | 19.39    |   71.78  | **71.78** |
>
> ---
>
> *Q7:* Other tasks
>
> *A7:* Adversarial noises have been demonstrated to be also effective in other tasks such as facial recognition in (Huang et al. 2021). Unfortunately, due to time constraints, we are not able to include more experiments on facial recognition. But we will conduct more experiments and add it to the camera-ready version.
>
> ---
> *Q8:* Control over Single Class
>
> *A8:* The accuracy of the model on locked ‘bird’ class alone is always below 5%.  We have added annotations to the heatmap plots for the single-class and multi-class learnability control experiments. Please see Figure 4 and Figure 5 in the appendix.
>
> Thanks again and further feedback is more than welcomed.

---

> ### Author Response · Authors · 2021-11-21
> **A friendly reminder of rebuttal conclusion**
>
> We’d like to express our gratitude for your detailed comments and constructive suggestions. We have responded to each of your questions and added clarifications to our revised manuscript. We hope they could be helpful addressing your concerns. In addition, we are more than happy to address any further questions before the conclusion of the rebuttal period.

---

> ### Author Response · Authors · 2021-11-27
> **Follow-up**
>
> Thanks again for your comments. We’d like to send a gentle reminder as the final deadline is approaching. Please let us know if you still have questions and we are happy to answer or discuss them.

---

### Official Review · Reviewer_ZNq5 · 2021-11-02

**Correctness:** 4
**Technical Novelty And Significance:** 3
**Empirical Novelty And Significance:** 3
**Recommendation:** 8
**Confidence:** 3

**Main Review:**

Strengths:
+ Interesting setting and a challenging threat model
+ Thorough evaluation

Weaknesses:
+ Narrative being very far from reality
+ Unclear contributions over and above Huang et al.


First, I want to mention that the paper is very well written, thoroughly evaluated and presents a convincing technical story for an ML audience. Yet, for people more familiar with Computer Security, and “do not roll your own crypto” principle, it presents a rather weak argument as to why any protection is provided with such an encryption scheme at all. In fact, the results from adversarial training tell that the system design is broken and the paper contradicts its own story.

Second, Carlini et al. in ‘Is Private Learning Possible with Instance Encoding?’ talk about lack of rigorous notions for privacy and use of ad-hoc arguments in instance encoding schemes such as InstaHide. It seems that this paper follows the same premise and provides no justifications for its, rather strong, claims. I am very confused about the setting of the paper.

```
Existing methods include training ML models on encrypted data, where the sensitive information could be hidden through cryptographic approaches in order to prevent malicious manipulation (Hesamifard et al., 2017; Ding et al., 2021). However, those kinds of data processing methods normally do not preserve visual properties from the raw data and thus affect normal use. For example, it does not make sense for one to send a fully encrypted and unrecognizable “selfie” photo to friends just to make sure the photo will not be exploited without authorization.
```

Indeed, if the goal is confidentiality and control of ones secrets, it does make sense to send data encrypted, a de-facto standard in private end-to-end communication. There is a branch of cryptography focusing on controlled disclosure of information if needed e.g. Partially Revealing Cryptography or even Deniable Cryptography. What is more, it does make sense to send ones ‘selfie’ encrypted over vetted channels and theoretically sound protocols — that way privacy is not going to be violated. In fact, the paper does discuss that data transfer is a potential security risk, yet argues to use a mechanism that leaks data by default:

```
From the data providers’ perspective, this means that any mistake conducted in the data transfer procedure could lead to potential privacy/security risks. This raises great challenges in securely distributing sensitive data from the data providers to the clients.
```

It took cryptographic community decades to establish best practices, develop theoretically and practically rigorous attacks and defences. Currently, there are international standards, competitions, and good understanding of privacy guarantees of different encryption mechanisms. ‘’Rolling your own crypto’’ leads to mistakes. Thus, I am unable to understand how the approach proposed in this paper improves upon the state-of-the-art in secure data transfer in Machine Learning.

In light of the above, I was unable to understand the threat model in which the paper proposes to provide a sensible mechanism to protect ones privacy, limiting its contributions over and above Huang et al. I also did not understand how the proposed approach innovates over a scheme constructed using standard cryptography toolbox like AES to make learning impossible.  If the authors are able to clarify these concerns, I would be happy to revise my review.







**Summary Of The Paper:**

Paper presents an idea of “Learnability lock”, a system that aims to control learnability of individual datapoints or data classes. The system builds on top of recent work on data learnability, where, in essence, noise is applied to individual datapoints in a way to disrupt learning generalisable features. From the description, it appears that it works as a poisoning or a backdoor attack, where the objective is to correlate non-generalisable features.

Now, learnability lock describes a mechanism that enables unlearnable features, whilst at the same time allowing the user to revert unlearnable noise if authorisation is acquired. Authors evaluate Learnability lock with a number of datasets and network architectures to find that it outperforms its competitors in presence of defences.




**Summary Of The Review:**

Paper presents an interesting construction for controlled learnability, but the setting of the paper seems to be a bit misleading. Paper argues that learnability lock provides privacy through their encryption scheme, yet in the same paper authors break their own scheme. Paper also provides no cryptographic underpinning as to why any privacy is provided at all.

---

> ### Author Response · Authors · 2021-11-18
> **Response to Reviewer ZNq5**
>
>
> Thank you for your valuable comments.
>
> *Q1:* Why any privacy protection is provided with such an encryption scheme
>
> *A1:* We believe there is a misunderstanding here and we are sorry for the confusion. We did **NOT** claim any certified protection of data privacy. In fact, we agree with you that the model trained based on our perturbed dataset can still leak private information (since our perturbation is essentially invisible). Instead, our purpose is to protect the **Intellectual Property** of data (as we mentioned in the abstract) by adding adversarial noises to the dataset so that models trained on the dataset would work badly. And the protection is attained when the model trained on our perturbed dataset is so bad that it won't be considered as a realistic inference model by the unauthorized attacker. Therefore, our aim is fundamentally different from privacy protection. To be more precise, our method aims to prevent unauthorized people from using the data while privacy protection considers the case when the data is already being used (to prevent sensitive information from being extracted). We hope this clears your concern on the correctness of our paper.
>
> We also want to further clarify the fundamental difference with traditional encryption methods or “instance encoding”. Both of them aim to preserve the accuracy of models trained on the encoded dataset, while hiding the information from the original dataset. In sharp contrast, our method aims to keep the visual features unchanged. As in the selfie example, uploading a fully encrypted “selfie” photo to social networks can certainly protect the photo from being exploited without authorization, yet it is against the user's original intention of sharing the photo to the public. Another example is that photographers like to share their work online and welcome others to buy the copyright for commercial uses. A fully encrypted photo would protect it from illegal exploitation but also kill the chances of selling copyrights.
>
> To summarize, our purpose is to protect the Intellectual Property of data, which is fundamentally different from traditional privacy protection or encryption methods. We have revised the paper to clarify these differences and we hope this will clear all the misunderstandings.
>
> ---
> *Q2:* Adversarial Training breaks the scheme
>
> *A2:* We respectfully disagree. In fact, the adversarial training experiments exactly showed our advantage over previous works as we achieved much lower accuracy to make it even more unlearnable. We also discussed in Appendix B.3 that use of adversarial training always leads to large degradation of performance compared with naturally trained models. This is extremely undesirable in an industrial setting and will make malicious attackers drop the interests of using such training data.
>
> ---
> *Q3:* Compared with standard cryptography toolbox like AES to make learning impossible
>
> *A3:* To the best of our knowledge, AES is a symmetric block cipher that aims to hide vulnerable information. To serve as, for instance, a good image encryption method in this sense, the encrypted image should be adequately masked so that nothing can be told about the original image. For the reasons we clarified in A1, it does not fit our purpose to protect the Intellectual Property of data while preserving the visual features of the dataset.
>
> Thanks again and further feedback is more than welcomed.

---

> > ### Comment · Reviewer_ZNq5 · 2021-11-22
> > **further comments**
> >
> > Thank you very much for your response.
> >
> > I am afraid the revised paper still has not addressed my concerns — I think that the abstract/introduction/conclusion are misleading. I do not think that use of the word encryption or encryption scheme is applicable here at all, and instead class-wise poisoning is more applicable. In fact, in the comments you claim IP, a digital rights management setup (DRM), whilst in the introduction of the paper you talk about encryption and protecting data ``in transit`` (?).  You will discover that most modern DRM works using standard cryptographic tools.
> >
> > As I mentioned before, there exist subfields of cryptography that enable the tasks you described e.g.``revealing encryption`` or ``deniable cryptography``, that both protect data and reveal some of its properties. If you follow down that path and want to claim secrecy, I think you need to consult the relevant literature.
> >
> > As I said before, if you change the setting of the paper i.e. relevant parts of abstract/intro/conclusion, I will change the score.

---

> > > ### Author Response · Authors · 2021-11-23
> > > **Revisions**
> > >
> > > Thank you so much for your further clarification and suggestions.
> > >
> > > We have revised the manuscript to avoid misleading statements. Please see our updated abstract/introduction/conclusion for the following revisions (marked in green color in our latest revision):
> > >
> > > - We removed contexts related to encryption, privacy, traditional cryptographic concepts as well as data transfer and IP protection that may cause confusions.
> > > - We formulated the problem of making datasets unlearnable as “learnability attack” and clearly stated its close relationship with traditional data poisoning attacks.
> > > - We clarified our scheme (in abstract, introduction and conclusion) as invertible learnability control based on the previous literature in learnability attack.
> > >
> > > We hope this will address your previous concerns and please also let us know if you have any other concerns or suggestions.

---

> > > > ### Comment · Reviewer_ZNq5 · 2021-11-29
> > > > **Response**
> > > >
> > > > Dear authors, thank you very much for your update, I think it clarifies my concern about the setting. Updating the score.
> > > >
> > > > Aside from that, I was wondering if there was a particular reason why W and B have to be learned? Aside from cases where the inverse does not exist, could they not be inverted directly?

---

> > > > > ### Author Response · Authors · 2021-11-29
> > > > > **Thank you & Response**
> > > > >
> > > > > Thanks for your recognition and increasing your score.
> > > > >
> > > > > In terms of your question on why we need to learn W and B, note that they are optimized for solving a min-min optimization problem (eq (3.1)). And the goal of this optimization is to make sure that the linear transformation formed by W and B can lead to “unlearnable” examples that can not be used to train useful machine learning models. In fact, if we apply random W and B to form the linear transformation, the generated dataset can still be learnable (is able to train meaningful machine learning models with fairly high accuracy). Therefore, the “learnability'' is our main concern, not the invertibility: without learning, some designed W and B can still be invertible but they do not satisfy the “unlearnable” requirement.
> > > > >
> > > > > We hope this answers your question and please let me know if you have any other questions.

---

> > > ### Author Response · Authors · 2021-11-27
> > > **Follow-up**
> > >
> > > Thank you again for all your suggestions to strengthen our paper. We’d like to remind you of our latest revisions as stated in our previous reply. Please let us know if you still have any concerns and we are pleased to discuss them before the final deadline.

---

> ### Author Response · Authors · 2021-11-21
> **A friendly reminder of rebuttal conclusion**
>
> We sincerely appreciate your detailed feedback on the mechanism of learnability lock. We have responded to each of your concerns and revised the manuscript accordingly to clear up confusions. Please let us know if your concerns have been addressed. If you have any further questions, we are more than happy to address them before the conclusion of the rebuttal phase.

---

### Decision · Program_Chairs · 2022-01-20

**Decision:**

Accept (Poster)

**Comment:**

The paper considers a relevant and interesting problem of protecting the intellectual property of data. The goal of the proposed method is to prevent unauthorized usage of the data, and the protection is attained when a model trained on the perturbed dataset will predict poorly and thus cannot be considered as a realistic inference model by the unauthorized attacker.

Technically, the paper tackles the problem of "unlearnable examples": to perturb the images of a labeled dataset to obtain perturbed dataset such that models trained on perturbed dataset have significantly lower performance, the perturbations are small, and one can approximately recover the original labeled dataset with the correct "secret key" (learnable parameters).

The authors propose two invertible transformations to craft adversarial perturbations: linear pixel-wise transformation and convolutional functional transformation based on invertible ResNet. Numerous experiments demonstrate the effectiveness of the proposed transformations in both securing the data (making the data unlearnable when transformation is applied) and unlocking the transformation (making the data learnable when the transformation is inverted).

The paper is well motivated and exhibits competitive results. Although there are some concerns about the similarity of the work compared with [1], we believe the additional constraint of this work, that one can approximately recover the original labeled dataset with the correct "secret key",  justifies a significant contribution.

[1] "Unlearnable Examples: Making Personal Data Unexploitable" Huang et al., ICLR '21